# Docs2Table: Bridging Multiple Documents and Structured Tables via LLMs

## Abstract

The task of Text-to-Table has garnered significant attention due to the critical value of structured data in information retrieval. Existing methods primarily focus on single-document scenarios, employing end-to-end generation to directly reproduce text fragments, while overlooking the practical need for multiple documents comparative analysis and the issue of domain variability. To address this, we introduce a novel information extraction task—multiple documents to table generation (**Docs2Table**), which requires models to comprehend the content of multiple documents, identify their similarities and differences, and generate structured tables tailored to domain-specific needs. We construct the first multi-domain benchmark dataset in Docs2Table, **FGLM**, covering finance, government, law, and medicine, with all data sourced from real-world business scenarios. Furthermore, we propose a two-stage pipeline method named **DDST** (Docs-Domain-Schema-Table) that introduces a domain gate selection mechanism, integrates domain-specific characteristics, and leverages JSON Schema as an intermediary to generate table headers and their value constraints, thereby improving table generation accuracy. Experimental results demonstrate that DDST achieves state-of-the-art performance on both traditional datasets and FGLM, significantly outperforming existing methods, with further analysis indicating strong generalization capabilities across different domains.

## 1 Submission of conference papers to ICLR 2026

Structured information extraction from unstructured text is fundamental in natural language processing (NLP), enabling various downstream applications. As an important subfield of structured information extraction, the text-to-table generation task, which involves summarizing complex information Wang et al. (2020) from lengthy documents into structured tables, is particularly valuable in domains requiring quantitative analysis, including business intelligence, legal documentation, and healthcare reporting. However, current approaches primarily focus on single-document scenarios Stapleton & et al. (2020), overlooking the critical ability to synthesize and compare information across multiple documents Gupta & et al. (2023); Fabbri et al. (2019) in real-world tasks such as financial analysis, policy comparison, and medical research.

Current Text-to-Table methods face three primary limitations. First, they rely on simplistic datasets derived from table-to-text tasks Bao et al. (2018); Lebret et al. (2016b); Parikh et al. (2020), where tables and texts exhibit near-isomorphic relationships Liu et al. (2021). These datasets lack the complexity of real-world scenarios, where tables must aggregate, reason over, and reconcile information from heterogeneous sources while capturing comparative insights. Second, existing methods emphasize end-to-end generation for single documents, failing to address the challenges of multi-document analysis, such as identifying contradictions, redundancies, and domain-specific patterns across diverse sources. Third, while large language models (LLMs) Brown & et al. (2020); Radford & et al. (2019); Devlin & et al. (2019) show promise in structured generation Jain & et al. (2024); Brown & et al. (2023), their ability to generate high-dimensional, domain-adapted tables, particularly in zero-shot or few-shot settings Ramu et al. (2024), remains underexplored.

To address these gaps, we introduce Docs2Table, a novel task that bridges multi-document analysis and structured table generation. Unlike traditional single-document text-to-table paradigms, Docs2Table requires models to: (1) synthesize information from heterogeneous documents Tang

Figure 1: An overview of the differences between the Docs-to-Table task and the previous Text-to-Table task.

et al. (2024), (2) discern similarities and differences across sources to enable comparative analysis, and (3) generate domain-specific tables that adhere to rigorous schema constraints. Docs2Table facilitates a global understanding of multiple documents by producing intuitive, structured tables that highlight comparative insights, addressing real-world needs in fields like computational social science and digital humanities where analyzing large sets of documents is common.

We introduce FGLM, the first multi-domain benchmark dataset covering Finance, Government, Legal, and Medical domains, curated from real-world cases. FGLM challenges models to process complex, lengthy texts and generate tables with integrated attributes. We propose DDST, a two-stage pipeline that uses domain gate selection to align documents with domain-specific schemas, followed by schema-guided table generation. By separating schema inference and content population, DDST improves fidelity and adaptability across domains. Experiments show DDST outperforms T3 and TKGT on FGLM.

Our contributions are summarized as follows:

- We propose Docs2Table, a novel task for multiple documents table generation, emphasizing cross-document synthesis and domain adaptation.

- We release FGLM, a multi-domain dataset with real-world complexity, addressing the scarcity of benchmarks for long-text, multiple documents table generation.

- We design DDST, a schema-driven pipeline that combines domain-aware schema inference with constrained generation, significantly improving accuracy and robustness in Docs2Table.

## 2 TASK DEFINITION

As shown in Figure 1, the Docs2Table task introduces a novel paradigm in information extraction, bridging the gap between unstructured text analysis and structured data generation.

| Dataset | DN | OT | TW | AW/D | TFW(%) | TF | TVTF |
|---|---|---|---|---|---|---|---|
| Wikitabletext | 13318 | Entity | 185111 | 13.90 | 50.04% | 2443 | 2262 / 791 / 1022 |
| Wikibio | 728221 | Entity | 70257683 | 96.48 | 45.22% | 2996 | 2771 / 1400 / 1406 |
| E2E | 51426 | Entity | 1152364 | 22.41 | 49.04% | 7 | 7 / 7 / 7 |
| Rotowire | 4853 | Event | 1637820 | 337.49 | 39.97% | 33 | 33 / 33 / 33 |
| CPL | 850 | Event | 1149207 | 1105.94 | 65.58% | 97 | 97 / 97 |
| LIVESUM | 3771 | Event | 4736376 | 1256 | - | 9 | 9 / - / 9 |
| FGLM | 1802 | Event | 17541539 | 9734.48 | 88.59% | 58 | 58 / 58 / 58 |

Table 1: Statistics of different datasets. DN represents the number of documents, OT represents the object type, TW represents the total words, AW/D represents the average words per document, TFW represents the table frequency weight, TF represents template frequency, and TVTF represents template-variant template frequency.

## 2.1 INPUT SPECIFICATION

The input to the Docs2Table task consists of a collection of related documents $D = \{d_1, d_2, \ldots, d_n\}$, where each document $d_i$ contains unstructured or semi-structured text. These documents are assumed to share a certain degree of semantic similarity or overlap Niu et al. (2018); Salton et al. (1975), ensuring that they collectively describe a relevant topic or event. The similarity among documents can be quantified using text similarity metrics, which ensures that the input documents are not entirely disjoint but rather form a cohesive corpus for analysis.

## 2.2 OUTPUT SPECIFICATION

The output of the Docs2Table task is a structured table $T$, a matrix where each row $r_j$ corresponds to a document $d_j$ in the input set $D$, representing distinct entities or objects. Each column $c_k$ represents an attribute derived from the documents, capturing key characteristics. The table schema $S$, comprising column headers and constraints, is dynamically inferred from the documents' content and structure.

Formally, the table $T$ is defined as:

$$T = \{r_1, r_2, \ldots, r_m\}, \quad \text{where } r_j = \{v_{j1}, v_{j2}, \ldots, v_{jk}\}$$

Here, $r_j$ is the $j$-th row for document $d_j$, and $v_{jk}$ is the value for attribute $c_k$, enabling intuitive comparisons across documents.

## 2.3 SIMILARITY CONSTRAINTS

To ensure document relevance in the Docs2Table task, we enforce similarity constraints to avoid sparsely filled tables lacking meaningful connections. We use **cosine similarity** Gusfield (1997); Singhal (2001) with **TF-IDF** Harper & Provost (2014); Jones (1972); Tata & Patel (2008) to measure document similarity. Each document is converted into a TF-IDF vector Arroyo-Fernández et al. (2017); Jalilifard et al. (2020), emphasizing key terms based on their frequency within a document and across the corpus. Cosine similarity between documents $d_i$ and $d_j$ is calculated as:

$$\text{sim}_{\cos}(d_i, d_j) = \frac{\mathbf{v}_i \cdot \mathbf{v}_j}{\|\mathbf{v}_i\| \|\mathbf{v}_j\|}$$

where $\mathbf{v}_i$ and $\mathbf{v}_j$ are TF-IDF vectors, and $\|\mathbf{v}_i\|$, $\|\mathbf{v}_j\|$ are their magnitudes. Values range from 0 to 1, with higher values indicating greater similarity. A **similarity threshold** $\tau_{\cos}$ ensures only documents with $\text{sim}_{\cos}(d_i, d_j) \geq \tau_{\cos}$ are included for analysis.

## 3 DATASETS AND STATISTICS

To address the limitations of existing datasets in capturing the complexity of real-world multiple documents scenarios, we introduce FGLM, a novel benchmark dataset for the Docs2Table task. As

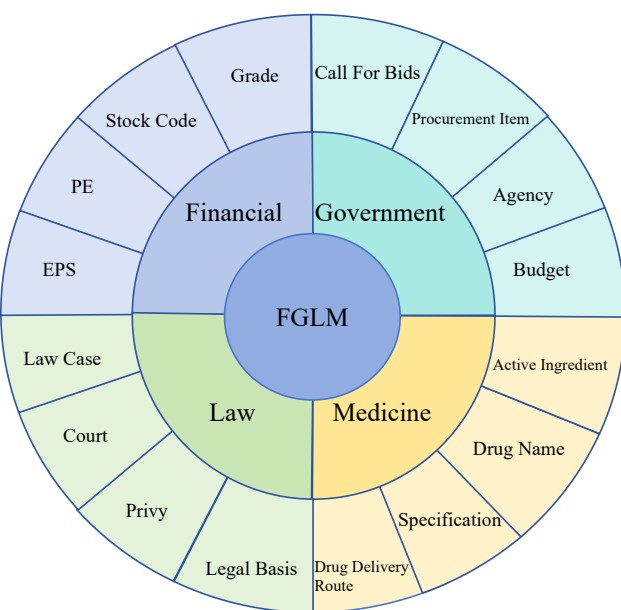

Figure 2: Four major domains that the Docs-to-Table task focuses on in the FGLM dataset (inner circle), as well as the common table field under each domain (outer circle).

shown in Figure 2, FGLM covers four key domains—finance, government, law, and medicine—all derived from authentic business cases. Each domain has been carefully selected to reflect the unique characteristics and challenges of extracting structured information from lengthy multiple documents inputs, as outlined in Appendix A.

## 3.1 DATASET COMPOSITION

FGLM contains 1,802 entries, with each entry comprising 10 documents. Each entry is associated with a single structured table that summarizes key information extracted from the corresponding documents. The dataset spans four key domains selected for their practical importance and unique challenges in structured information extraction.The dataset details are in Appendix B.To ensure dataset quality and consistency, we conducted extensive cleaning. Raw documents and tables were collected from authentic sources. Short texts were filtered out to focus on long documents critical for multiple documents analysis, while stop words and POS tags were removed, retaining core content (88% of total). Table fields not present in the documents were deleted to ensure alignment. For every 10 documents, benchmark tables were constructed by summarizing the document and the original content, ensuring an accurate representation.Furthermore, based on our observation of the necessity and prevalence of multiple documents to table information extraction in real-world business scenarios, our documents and tables are obtained from real-world websites and corrected by domain experts, resulting in gold tables that are informative and interpretable.

## 3.2 STATISTICAL PROPERTIES

As shown in Table 1, the FGLM dataset, designed for the Docs-to-Table task, features an average document length of 9,734.48 tokens, significantly longer than other benchmark datasets, posing challenges in processing and integrating unstructured information. Its output tables average 14.5 dynamically inferred fields, requiring complex structured outputs. With 1,802 entries across Finance, Government, Legal, and Medical domains, FGLM enhances domain generalization and provides a robust benchmark for multi-dimensional semantic scenarios. Unlike datasets from Table-to-Text tasks (e.g., Wikitabletext Wiseman et al. (2017), Wikibio, E2E Novikova et al. (2017), Rotowire Le-

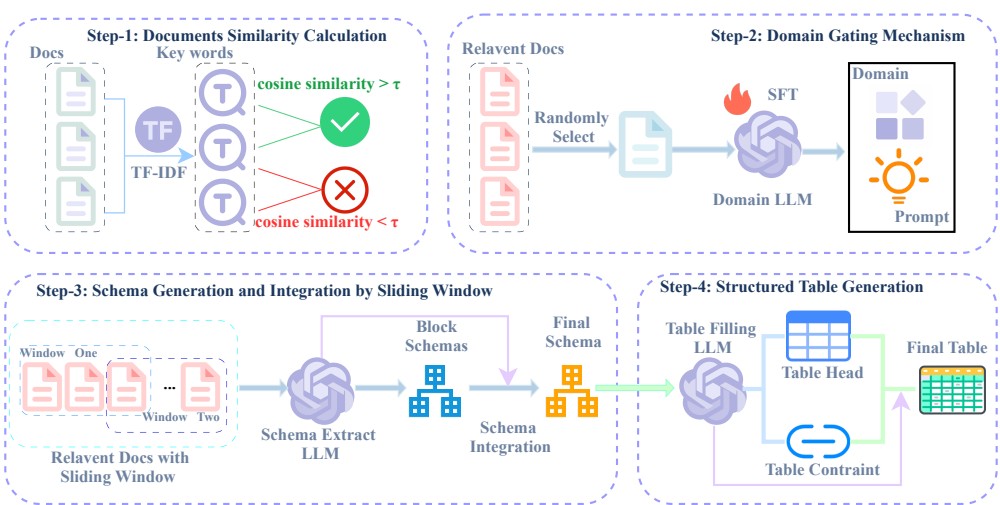

Figure 3: Overview of pipeline of DDST.

bret et al. (2016a)) or existing Text-to-Table datasets (e.g., CPL Deng et al. (2024), LIVESUM Jiang et al. (2024)), FGLM is the first to focus on multi-document input for structured table generation. Its dynamic field generation and cross-paragraph information fusion make it a realistic platform for evaluating advanced information extraction and structured generation models, with 58 core fields supporting end-to-end data flow across training, validation, and testing.

# 4 DDST(DOCS-DOMAIN-SCHEMA-TABLE) TWO-STAGES PIPELINE

To tackle multi-document table synthesis with domain adaptability, we propose **DDST**, a two-stage pipeline combining domain-aware schema induction Zhang & et al. (2023); Li & et al. (2024) and constrained table generation. As shown in Figure 3, DDST uses a domain gate selection mechanism to detect domain-specific patterns, dynamically infers table schemas, and employs model retrieval to populate tables under schema constraints. The methodology includes three steps: domain gating, schema generation and integration, and structured table generation.

## 4.1 DOMAIN GATING MECHANISM

Inspired by the Mixture-of-Experts (MoE)Jacobs et al. (1991) architecture, we introduce a domain-gating mechanism that selects domain-specific prompts. The key idea is that different domains exhibit distinct terminologies and table schema expectations. Specifically, we randomly select two out of the original ten documents and use a large language model (LLM) to determine the domain of this document group. Based on the identified domain, the model generates a domain-specific prompt and a list of salient domain-specific keywords. These elements are injected into the initial instruction prompt to guide the model in schema construction. This enriched prompt serves two purposes: it reflects the domain's semantic focus, and it enhances the relevance and completeness of the schema fields in the next step.

## 4.2 SCHEMA GENERATION AND INTEGRATION

The second stage generates a JSON schema Bourhis et al. (2017) to define the target table's structure. To manage long documents Chen & et al. (2023), we use a sliding window mechanism Gou et al. (2020), dividing documents into overlapping windows (minimum 2 documents, step size smaller than window size) to capture inter-document relationships. For each window, the LLM creates a preliminary JSON schema with property names, data types, descriptions, and constraints. Schemas are refined iteratively by promoting common attributes to top-level properties, ensuring consistency

and minimizing headers. The final schema, including fields like subject, type, description, and constraints, guides structured table generation Wang & et al. (2023); Gao & et al. (2023).

## 4.3 STRUCTURED TABLE GENERATION

In the structured table generation phase, we use a unified JSON Schema's field definitions -to guide information extraction. Each document is treated as an instance, and the model extracts values for schema-defined fields while enforcing data types, required fields, and value ranges. Prompts explicitly instruct the model to follow schema rules for accuracy and consistency. The final output is a column-oriented JSON format, where each key represents a table column with an array of values extracted across all documents. This aligns with standard tabular structures and facilitates other nlp tasks.

## 5 EXPERIMENTS

This chapter provides a detailed exposition of the experimental design, datasets, models, evaluation methodologies, and results analysis for the Docs2Table task. The experiments are designed to address the following key research questions:

| Models | First Column F1 | | | Table Header F1 | | | Data Cell F1 | | | | |
|---|---|---|---|---|---|---|---|---|---|---|---|
| | Exact | Chrf | BS | Exact | Chrf | BS | Exact | Chrf | BS | ERR | EHR |
| *Fine-Tune* | | | | | | | | | | | |
| Mistral-7B-Instruct-v0.3 | **11.12** | **16.36** | **17.18** | 3.72 | **8.39** | **13.17** | 7.52 | **10.51** | **11.91** | **6.54** | 20.59 |
| TableLLM-7B | 10.24 | 14.21 | 15.64 | 2.89 | 5.81 | 10.24 | 5.67 | 8.83 | 9.71 | 7.08 | 24.60 |
| StructLM-7B | 7.89 | 12.77 | 13.46 | 3.11 | 6.53 | 10.56 | 6.01 | 8.66 | 10.37 | 7.30 | **28.70** |
| *Zero-Shot* | | | | | | | | | | | |
| Qwen2.5-72B | 27.99 | 38.78 | 37.45 | 8.27 | 19.07 | 27.01 | **18.86** | 25.81 | 27.74 | 4.54 | 69.46 |
| Qwen2.5-72B (CoT) | **28.95** | **40.02** | 39.01 | 7.48 | 18.03 | 28.04 | 17.02 | 25.03 | 27.01 | 4.01 | 71.02 |
| DeepSeekV3 | 25.33 | 32.65 | 35.54 | **9.82** | 19.49 | 29.56 | 17.08 | 26.21 | 30.84 | 4.40 | 43.28 |
| DeepSeekV3 (CoT) | 25.97 | 33.01 | 36.02 | 9.01 | 18.02 | 30.03 | 16.01 | 25.02 | 30.01 | 4.00 | 44.01 |
| GPT-4o | 21.85 | 28.08 | 28.73 | 8.55 | 19.56 | 26.88 | 14.82 | 27.66 | 28.34 | 4.10 | 54.39 |
| GPT-4o (CoT) | 22.01 | 28.03 | 29.01 | 8.02 | 20.01 | 27.02 | 14.01 | 27.78 | 29.31 | **3.51** | 55.02 |
| Llama-3.3-70B | 26.79 | 37.37 | 40.52 | 7.48 | 18.35 | 27.61 | 17.83 | 24.81 | 27.51 | 4.83 | 68.42 |
| Llama-3.3-70B (CoT) | 27.01 | 38.02 | **41.01** | 7.01 | 17.02 | 28.03 | 17.01 | 24.02 | 28.11 | 4.51 | 69.01 |
| Qwen3-32B | 24.66 | 31.42 | 35.07 | 9.18 | 23.75 | 30.14 | 15.56 | 28.63 | 33.14 | 3.99 | 80.86 |
| Qwen3-32B (CoT) | 25.01 | 32.01 | 36.02 | 8.51 | **24.01** | **31.02** | 15.01 | **30.12** | **35.67** | **3.51** | **82.01** |
| *Zero-Shot with DDST* | | | | | | | | | | | |
| Qwen2.5-72B(DDST) | 31.90 | 44.07 | 45.32 | 13.00 | 28.71 | 42.38 | 21.40 | 39.24 | 41.30 | 4.59 | 76.23 |
| DeepSeekV3(DDST) | 30.85 | 38.47 | 39.90 | **15.06** | **30.79** | 48.24 | 17.45 | 36.65 | 45.60 | 5.79 | 56.58 |
| GPT-4o(DDST) | 21.00 | 22.12 | 21.38 | 7.93 | 29.21 | 47.74 | 22.22 | 37.79 | 46.97 | 5.21 | **79.86** |
| Llama-3.3-70B(DDST) | 30.92 | 42.76 | 45.06 | 10.52 | 25.87 | 42.21 | 20.33 | 38.22 | 42.60 | 4.65 | 72.85 |
| Qwen3-32B(DDST) | **32.96** | **48.12** | **49.10** | 12.86 | 30.54 | **48.46** | **22.63** | 39.28 | 49.58 | **4.23** | 68.72 |
| *Thinking LLMs* | | | | | | | | | | | |
| DeepSeek-r1 | 35.21 | 41.28 | 43.56 | 42.39 | **52.34** | **56.78** | 47.83 | 52.39 | 56.78 | 2.87 | 85.63 |
| o4-mini | **38.76** | 39.87 | 41.34 | 38.76 | 45.63 | 48.92 | 45.21 | 50.12 | 54.32 | **2.34** | **89.76** |
| Grok3 | 33.45 | 40.92 | 42.34 | **45.63** | 48.17 | 48.17 | **51.24** | **57.89** | **62.34** | 3.21 | 83.42 |
| Gemini 2.5 Pro | 34.56 | **45.63** | 47.89 | 40.12 | 47.89 | 50.12 | 46.78 | 51.23 | 55.67 | 3.12 | 84.56 |
| *Thinking LLMs with DDST* | | | | | | | | | | | |
| DeepSeek-r1(DDST) | 37.65 | 44.56 | **51.23** | 44.13 | 57.14 | 57.01 | 51.15 | 59.87 | 66.12 | 1.38 | **91.12** |
| o4-mini(DDST) | 38.88 | 43.54 | 45.13 | 47.31 | **60.19** | **60.23** | 51.78 | 60.47 | 67.45 | **1.21** | 90.22 |
| Grok3(DDST) | **39.12** | 43.32 | 43.42 | **48.94** | 58.32 | 59.11 | 53.78 | **62.75** | **70.34** | 1.47 | 88.31 |
| Gemini 2.5 Pro(DDST) | 36.15 | **47.18** | 47.98 | 45.15 | 55.15 | 57.68 | **54.14** | 59.11 | 63.15 | 1.59 | 85.64 |

Table 2: Results of Fine-Tune LLMs, state-of-the-art Non-Thinking LLMs in zero-shot, CoT, Thinking LLMs, and our DDST settings on FGLM, showing average metrics with maximum values in bold, second maximum values underlined, and minimum ERR values in bold, second minimum ERR values underlined.

1. How do existing fine-tuned models, state-of-the-art LLMs, thinking LLMs, and models integrated with the DDST method perform on the FGLM dataset?

2. How does the DDST method enhance model performance, and what are the contributions of its individual components?

3. How does the DDST method compare with other methods across different domains and evaluation metrics?

## 5.1 SETUP

**Baseline Models**  The experimental models are divided into five groups to evaluate their performance on the Docs2Table task. The first group consists of fine-tuned models, including Mistral-7B-Instruct-v0.3 Jiang & et al. (2023), TableLLM-7B Tang & et al. (2023), and StructLM-7B Wang & et al. (2023), which have been optimized using state-of-the-art fine-tuning techniques on tabular datasets to enhance information extraction and table generation capabilities. The second group comprises state-of-the-art LLMs in a zero-shot setting, including Qwen2.5-72B Team & Cloud (2024), DeepSeekV3 Team & AI (2025), GPT-4o OpenAI (2024), Llama-3.3-70B Touvron & et al. (2023), and Qwen3-32B Team & Cloud (2025), evaluated under a zero-shot setting with two prompting strategies: (1) direct instruction prompting, providing task descriptions and input text; and (2) chain-of-thought (CoT) prompting, incorporating "step-by-step reasoning" instructions to enhance reasoning capabilities. The third group includes zero-shot models integrated with DDST, used to compare the performance differences between the DDST approach and standard models. The fourth rooftop Thinking LLMs, including DeepSeek-r1, o4-mini, Grok3, and Gemini 2.5 Pro, which are renowned for their enhanced reasoning abilities. The fifth group, Thinking LLMs with DDST, evaluates the further optimization effects of the DDST approach on reasoning-enhanced models.

**Evaluation Metric**  To comprehensively evaluate table generation performance, we adopt the metric framework defined by Wu et al. (2021), encompassing three key aspects:

**First Column F1**  This metric assesses the model's ability to extract key entities from consolidated multi-document text using Exact Match, Chrf, and BERT Score. It reflects the model's capacity to identify core themes across documents.

**Table Header F1**  This evaluates the quality of generated table headers which measuring the model's ability to accurately represent key attributes across documents. Header quality is essential for understanding structural and differential information among documents.

**Data Cell F1**  This assesses the accuracy of content populated under the headers, reflecting the model's capability to search and fill data from multiple documents, serving as a key indicator of overall table quality.

Additionally, we introduce the following two metrics to evaluate the quality of header generation:

**Extra Header Rate (EHR)**  This measures the proportion of generated headers with data coverage exceeding 80%, allowing for a more flexible evaluation, as real-world tables may include fields beyond the ground truth.

**Error Rate (ERROR)**  This indicates the proportion of generated headers with data coverage below 20%, highlighting the failure rate and providing insight into the model's error rate.

## 5.2 EVALUATION ON THE FGLM DATASET

**Fine-Tune LLMs evaluation**  We compared model performance on the FGLM dataset using multiple metrics. Table 2 shows that Mistral-7B-Instruct-v0.3 outperformed TableLLM-7B and StructLM-7B among fine-tuned models. However, these models had error rates of 6.54%-7.30% and extra header generation rates of 20.59%-28.70%, indicating limitations in multiple documents integration and complex header generation. Analysis suggests insufficient fine-tuning data for multiple documents scenarios, and fewer parameters may cause detail loss and inflexible header generation.

**Non-Thinking LLMs evaluation**   Non-Thinking LLMs outperformed fine-tuned models, especially with the DDST method, demonstrating superior metrics across evaluations. Chain-of-Thought (CoT) prompts further boosted performance: First Column F1 exact match increased by 1%-2%, and Table Header F1 Chrf improved by 0.5%-1%, highlighting enhanced reasoning. Zero-shot models showed the lowest error rate (3.51%) but highest extra header generation (82.01%), reflecting LLMs' strong generative capacity. Notably, GPT-4o saw the largest DDST gains, with BERTScore-based Data Cell F1 rising from 28.34% to 46.97%, though it underperformed on First Column F1, likely due to reduced sensitivity to domain-specific content in core fields.

**Thinking LLMs evaluation**   Thinking LLMs outperform standard LLMs in the Docs2Table task due to their advanced reasoning capabilities. As shown in Table 2, Grok3 achieves the highest Data Cell F1 (BS) score of 62.34%, improving to 70.34% with DDST, surpassing standard LLMs like Qwen3-32B (CoT) at 35.67% and 31.02%. The o4-mini model excels with a 2.34% ERROR rate and 89.76% EHR, reflecting strong entity extraction and table generation. With DDST, o4-mini's ERROR rate drops to 1.21%, and DeepSeek-r1's EHR rises to 91.12%. DDST's domain gating and schema extraction significantly boost reasoning-focused models' performance, enhancing multi-document integration and reducing errors.

## 5.3   ABLATION STUDY

| Model | First Col F1 | | | Table Header F1 | | | Data Cell F1 | | | ERR | EHR |
|---|---|---|---|---|---|---|---|---|---|---|---|
| | E | C | BS | E | C | BS | E | C | BS | | |
| GPT-4o | 0.2185 | 0.2808 | 0.2873 | 0.0855 | 0.1956 | 0.2688 | 0.1482 | 0.2766 | 0.2834 | 0.0410 | 0.5439 |
| w/T3 | **0.2473** | **0.3147** | **0.3211** | 0.0614 | 0.2346 | 0.2518 | 0.1742 | 0.3147 | **0.3749** | **0.0143** | 0.7415 |
| w/DST | 0.2235 | 0.2795 | 0.2913 | 0.0231 | 0.1235 | 0.1568 | 0.0531 | 0.1127 | 0.1874 | 0.0712 | 0.7124 |
| w/DDSHT | 0.1035 | 0.1547 | 0.1634 | **0.0861** | 0.2132 | 0.2673 | 0.0311 | 0.0814 | 0.1029 | 0.0433 | 0.5871 |
| w/DDST | 0.2094 | 0.2212 | 0.2138 | 0.0793 | **0.2921** | **0.4774** | **0.2222** | **0.3779** | 0.4697 | 0.0521 | **0.7986** |

Table 3: Ablation study of prompting methods.   Best and second-best results are **bold** and underlined, respectively.

The ablation study uses GPT-4o to analyze the components of DDST by comparing: the baseline GPT-4o model; T3 (Text-Tuple-Table) (Tang & et al., 2023); DST (Docs-Schema-Table) without domain gate selection; DDSHT (Docs-Domain-Schema-Hybrid Retriever-Table), which uses hybrid RAG technology to retrieve relevant data from raw documents for table filling based on generated schemas, the second stage of the TGKT method; and DDST (Docs-Domain-Schema-Table).

Experimental results from Tab 3 show that the DDST method outperforms other methods across multiple metrics. The T3 method also performs well in specific indicators, which demonstrates its advantages in data cell generation. Additionally, the T3 method has an ERROR RATE lower than DDST's, indicating stronger capabilities in controlling erroneous header generation. The performance degradation of DST confirms the importance of the domain gate mechanism. DDSHT performs poorly, likely because generated header fields are typically condensed vocabulary from documents that cannot be directly matched in raw texts. When using RAG for header field matching, the method fails to retrieve relevant text chunks, significantly reducing its effectiveness.

## 5.4   THE COMPARISON BETWEEN T3 AND DDST

As shown in Figure 4, we extend the AUTO-QA method Jain et al. (2024) to generate QA pairs for evaluating table generation across five metrics: **coverage**, **format**, **reasoning**, **comprehension**, and **hallucination control** Fan & et al. (2023); Song & et al. (2023). We compare GPT-4o, GPT-4o (T3), and GPT-4o (DDST). **GPT-4o (DDST)** excels in coverage, reasoning, and hallucination control, but slightly trails T3 in format and comprehension. T3 leverages single-document triplet extraction for fine-grained tables, suitable for complex entity extraction but limited in multi-document connectivity for the Docs2Table task. In contrast, DDST uses a domain gating mechanism to identify domain-specific features and a sliding window approach for intermediate schema generation, enhancing cross-document connections. **DDST** demonstrates superior performance by covering more document information through domain gate selection and JSON Schema, especially excelling in

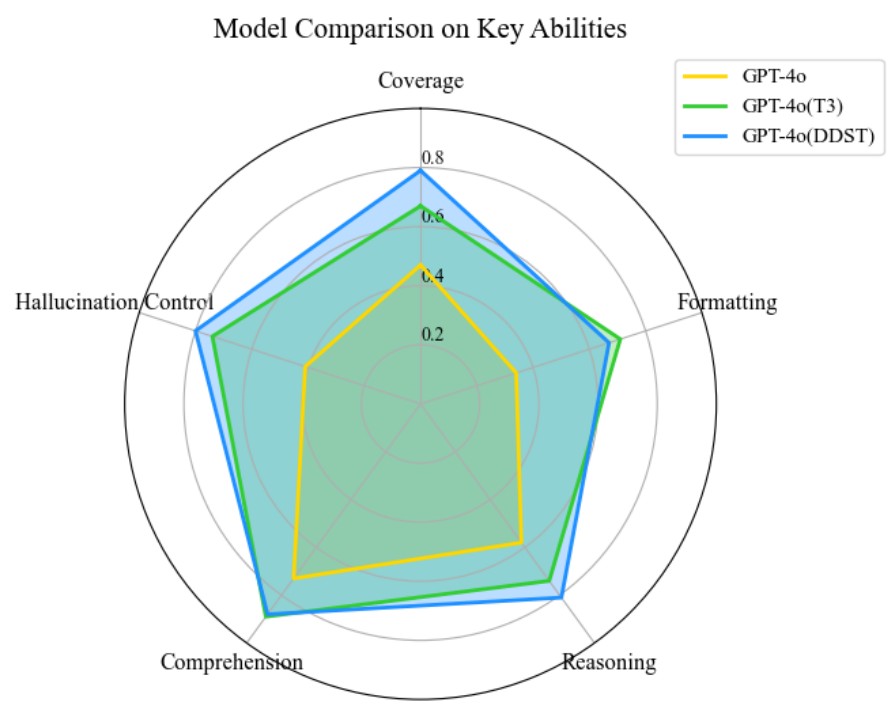

Figure 4: Visualization of the comparison between T3 and DDST.

financial and government affairs domains where tables contain key indicators and cross-document association information, enabling easier model reasoning and reduced hallucinations. Detailed information in Appendix C.

## 6  CONCLUSION

This paper introduces Docs2Table, a novel task for generating structured tables from multiple documents, and presents FGLM, the first real-world, multi-domain dataset spanning Finance, Government, Legal, and Medical domains, curated to challenge models with lengthy, complex texts. To address limitations in structured output and cross-document integration, we propose DDST, a two-stage pipeline that leverages a domain gate mechanism and JSON Schema as an intermediate representation to dynamically align documents with domain-specific schemas and enforce structural constraints. Experimental results demonstrate that DDST, when applied to state-of-the-art LLMs, outperforms traditional models and zero-shot methods across multiple metrics, particularly in complex domains like Legal and Government, where it excels in capturing key indicators and associations. Ablation studies confirm that domain knowledge enhances structured generation, while RAG negatively impacts table filling in FGLM. AUTO-QA analysis further highlights DDST's superiority in coverage, reasoning, and hallucination control, underscoring its effectiveness in structured information extraction from diverse, real-world documents.

ETHICS STATEMENT

The FGLM dataset and experiments in this work adhere to ethical research practices, utilizing publicly available sources from finance, government, law, and medicine domains, including financial reports, procurement announcements, court rulings, and pharmaceutical specifications, all compliant with transparency requirements. Personal information in legal and gov datasets was randomly replaced using named entity recognition (NER) techniques to ensure privacy. Data preprocessing maintained the integrity of the original sources, and experiments aligned with their intended research purpose of advancing multiple documents table generation. The dataset and code are open-sourced to promote transparency and, to the best of our knowledge, this work introduces no additional ethical risks or harms, adhering to data privacy and research ethics standards.

ACKNOWLEDGMENTS

We gratefully acknowledge the anonymous reviewers for their insightful comments and valuable suggestions.

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

## A    APPENDIX

## B    DETAILS OF FGLM DATASET

FGLM contains 1,802 entries, with each entry comprising 10 documents. Each entry is associated with a single structured table that summarizes key information extracted from the corresponding documents. The dataset is divided into four domains:

1. **Finance**: This domain contains 302 entries derived from public financial analysis reports of listed companies. Each report includes essential fields such as stock codes, report titles, rating changes, revenue forecasts, and other key financial indicators. These reports are designed to support investment decisions.

2. **Government**: With 500 entries, this domain focuses on open government procurement announcements. Each document contains information about procurement items, bidding information, project specifications, and agency details. These documents are crucial for understanding public sector expenditures and regulatory compliance.

3. **Law**: This domain contains 500 entries covering court rulings from public legal archives. These documents record detailed case information, including involved parties, legal basis, judicial decisions, and citations of relevant laws. They are vital for legal analysis and precedent research.

4. **Medicine**: This domain contains 500 entries detailing specifications of pharmaceutical products. Each document provides comprehensive information about drug formulations, administration routes, approval dates, and manufacturing details. These documents are essential for compliance monitoring and medical research.

## C    STRUCTURE OF FGLM JUDGEMENT

In this chapter, we will show the articles corresponding to each field of data. Since the length of each data article is very large, we will only intercept one of the data for display. Each data in the data set is stored in Chinese jsonl format.

FGLM examines four key datasets to provide a comprehensive analysis of market performance and investment opportunities. The first dataset , as shown in Figure 5 and  6, consists of stock information from 10 companies across various industries, including tourism, steel, semiconductors, and healthcare. It includes essential financial metrics such as stock codes, brief company descriptions, investment recommendations, and earnings forecasts for 2024 and 2025. These insights allow for a deeper understanding of market trends and can help investors assess the future potential of these companies based on their P/E ratios and growth expectations.

The second dataset, as shown in Figure 12 and  13, focuses on procurement project announcements from government and educational sectors. It details the names, procurement units, project budgets, and other vital information such as announcement times and project contacts. This data provides a window into public sector investments, offering potential market opportunities in industries related to infrastructure, technology, and public services. By analyzing these procurement details, stakeholders can identify upcoming market shifts driven by governmental spending.

The third dataset, as shown in Figure 14 and  15, involves legal case information, including case numbers, parties involved, case types, ruling dates, and outcomes. These details help track the evolution of legal precedents and outcomes, offering valuable insights into the trends and shifts in judicial decisions. The dataset enables a comprehensive analysis of the legal environment and can be instrumental in understanding the broader implications of legislative changes on business and investment strategies.

The fourth dataset, as shown in Figure 16 and  17, pertains to the healthcare sector, with a focus on pharmaceutical companies and their ongoing projects. It includes data on drug approvals, clinical trial phases, investment opportunities, and market forecasts. With growing interest in medical research and pharmaceutical innovations, this dataset offers insights into the potential breakthroughs and emerging trends that could reshape the healthcare landscape. It is especially relevant for in-



## Financial analysis reports of listed companies

Title: All Prerequisites for the Tender Offer Have Been Met

Information Source: www.eastmoney.com

Institution: TF Securities

Analyst: Sun Haiyang

Content:

O.R.G. Packaging Group Co., Ltd. (Stock Code: 002701)

All prerequisites for the tender offer have been met.

On December 13, 2024, Huarui Fengquan Co., Ltd., a subsidiary of the company, obtained the "Business Registration Certificate" related to the foreign exchange registration for this tender offer transaction, completing the required foreign exchange registration procedures. As of the disclosure date on December 13, all conditions precedent to the offer have been fulfilled.

Previously, the company intended to launch a voluntary conditional general offer to all shareholders of COFCO Packaging Holdings Co., Ltd., a Hong Kong-listed company, through its overseas subsidiary Beijing Huarui Fengquan Management Consulting Co., Ltd. (referred to as the "Offeror"), to acquire all issued shares of COFCO Packaging in cash. Currently, the company indirectly holds 24.40% of the shares in COFCO Packaging, which is a participating shareholder of the company. Upon completion of this transaction, the company expects to gain control over the target company.

If this integration is completed, coupled with an expected contraction in capital expenditures by leading companies, the industry supply-demand relationship may ease, and earnings and valuation levels of leading firms are expected to reach a bottom.

Through this acquisition, the listed company will strengthen its core business, expand strategic customer relationships, and promote long-term sustainable development. The company will further consolidate its main operations in two-piece cans and three-piece beverage and milk powder cans. It also plans to enrich its domestic product lines, including steel drums, aerosol cans, and plastic packaging, cultivating new profit growth drivers. Additionally, the company expects to achieve synergies with COFCO Packaging across multiple dimensions such as technology, marketing, production capacity, and supply chain. Through differentiated positioning, it aims to serve diverse customer needs, broaden strategic partnerships, reduce reliance on single customers, and support sustainable growth.

We believe that policies expected in 2025 will clearly stimulate consumption, benefiting the metal packaging industry as an upstream sector. Following deep industry integration, competitive dynamics are anticipated to be reshaped. The company maintains a stable customer structure and actively pursues a differentiated can-type strategy, positioning it well for future cooperation with COFCO in both customers and products. Moreover, domestic production lines are expected to undergo further consolidation and overseas expansion, offering broad medium- to long-term growth potential.

Domestic demand for two-piece metal cans remains stable, and in the medium term, the company is expected to benefit from an increase in beer canning rates. If this integration is completed, combined with a projected decline in leading companies' capital expenditures, the industry's supply-demand balance may improve, and earnings and valuation centers for leading players are expected to stabilize.

The company is the first A-share listed enterprise in China's metal packaging industry and has made notable progress in terms of business layout integrity, scale, and customer structure. It holds a leading position domestically in structural advantages, innovation capability, profitability, and comprehensive service capabilities. Furthermore, based on its industry-leading revenue scale, the company continues to make progress in new material applications, new can development, new service model expansion, and metal packaging recycling and reuse.

We forecast the company's operating revenues for 2024–2026 at RMB 14.19 billion / 15.01 billion / 16.39 billion, respectively, with net profits attributable to parent shareholders estimated at RMB 870 million / 970 million / 1.09 billion yuan. Earnings per share (EPS) are projected at RMB 0.34 / 0.38 / 0.43 per share, corresponding to price-to-earnings ratios of 18.05x and 16.15x for 2024 and 2025, respectively. We maintain a "Buy" rating.

Risk Warnings: Risks include underperformance of mergers and acquisitions, weaker-than-expected consumer demand, lower-than-expected pricing, and potential loss of key executives.



Figure 5: Financial analysis reports of listed companies.

vestors seeking to capitalize on new treatments, technologies, and regulatory changes within the pharmaceutical industry.

By combining insights from four datasets, the paper provides a holistic approach to forecasting future market trends. It identifies promising growth sectors, such as semiconductors and healthcare, alongside significant procurement projects across education, environment, and urban development. This combination of stock market performance and procurement data presents a valuable tool for investors seeking to optimize asset allocation and identify emerging opportunities in the coming years.

# D   DETAIL OF AUTO-QA EXPERIMENTS

We extend the AUTO-QA framework Jain et al. (2024) to evaluate table generation across five dimensions: coverage, formatting, reasoning, comprehension, and hallucination control. The experiments compare three methods: vanilla GPT-4o, GPT-4o(T3), and our proposed GPT-4o(DDST).

## D.1   QA PAIR GENERATION WITH AUTO-QA

To generate QA pairs, we use the AUTO-QA method, which relies on large language models, specifically GPT-4.1, for extracting document-related information and converting it into structured QA pairs. We define the following approach for QA pair generation:

| No. | Stock Code | Stock Name | Report Title | Eastmoney Rating | Rating Change | Institution | 2024 EPS Forecast (RMB/share) | 2024 P/E Ratio | 2025 EPS Forecast (RMB/share) | 2025 P/E Ratio | Industry | Date |
|-----|-----------|-----------|-------------|-----------------|--------------|-------------|------------------------------|---------------|-------------------------------|----------------|----------|------|
| 2 | 600754 | Jinjiang Hotels | Acquisition of minority stakes in core brands to drive growth; share buyback plan expected to boost market confidence | Overweight | Maintained | Guosen Securities | 1.13 | 23.8 | 1.29 | 21 | Hotel & Tourism | 2024/12/26 |
| 3 | 2075 | Shagang Steel | Profit temporarily pressured; "Special Steel + Gear" dual drivers ensure risk resilience | Overweight | Maintained | Dongwu Securities | 0.05 | 134.94 | 0.07 | 110.45 | Steel Industry | 2024/12/26 |
| 4 | 688620 | Angstrem Micro | Focused on IoT smart hardware SoC chips; continuous iteration and upgrading of technology platform | Overweight | Initiated | Huajin Securities | -0.07 | -157 | 0.19 | 61.2 | Semiconductor | 2024/12/26 |
| 5 | 300703 | Chuangyuan Shares | Solid fundamentals; expecting shareholder support | Buy | Initiated | TF Securities | 0.53 | 32.4 | 0.69 | 24.99 | Consumer Goods | 2024/12/26 |
| 6 | 688710 | InnoStar Bio | In-depth report: leading domestic player in non-clinical safety evaluation | Overweight | Initiated | Dongguan Securities | 1.16 | 34.01 | 1.26 | 31.19 | Biotechnology | 2024/12/26 |
| 7 | 603166 | Fuda Parts | First coverage report: new energy hybrid drives growth; three growth curves enter EV era | Buy | Initiated | Cinda Securities | 0.28 | 26.34 | 0.41 | 17.98 | Auto Components | 2024/12/26 |
| 8 | 426 | Xingye Silver | Proposed acquisition of 85% stake in Yubang Mining; entering global silver giants | Buy | Maintained | China Post Securities | 0.92 | 14.01 | 1.2 | 10.72 | Nonferrous Metals | 2024/12/26 |
| 9 | 600998 | Jiuzhou Tong | Commentary report: first medical REIT approved; asset structure optimization | Buy | Maintained | China Galaxy Securities | 0.44 | 11.86 | 0.51 | 10.27 | Healthcare Distribution | 2024/12/26 |
| 10 | 3021 | Zhaowei Electromechanical | Deep expertise in micro-drive systems; expanding into automotive electronics | XR | robotics | Overweight | 0.9 | 80.93 | 1.09 | 66.54 | Motors | 2024/12/26 |
| 11 | 1308 | Konka Tech | Leading player in smart display; launching innovative products to build third growth curve | Hold | Initiated | Huafu Securities | 1.25 | 22 | 1.64 | 17 | Consumer Electronics | 2024/12/26 |

Figure 6: Financial Table.

- **Prompt Design:** For each domain, a set of prompts is created to instruct the model to extract relevant information and convert it into question-answer pairs. These prompts are designed to focus on key aspects such as entity extraction, document information, and logical reasoning.

- **Model Selection:** The model used for QA generation is GPT-4.1, known for its large-scale understanding of natural language and document structure. The model generates responses based on the input documents in various domains, including finance, government, law, and medical data.

- **Evaluation of Five Dimensions:** Each QA pair generated by GPT-4.1 is evaluated on five dimensions:

  1. **Coverage:** The extent to which the QA pair covers key information in the document.

  2. **Format:** The adherence to a standardized table format.

  3. **Reasoning:** The model's ability to logically reason from the provided document.

  4. **Comprehension:** The depth of comprehension demonstrated by the model when answering questions.

  5. **Hallucination Control:** The ability to avoid generating incorrect or irrelevant information.



# Schema Extraction Prompt

"""

{domain_prompt}

Article content:

{articles}

Please design a suitable JSON Schema to extract the key information of the article, requirements:

1. The Schema should be able to contain the common information of all articles, and ensure that all generated attribute fields can be found in each article. For example, if attribute 1 can be found in the first article but not in the second article, then the attribute is discarded

2. Generate a corresponding description for each attribute field

3. Each attribute is a first-level title, and attributes involving time should be expressed separately

4. The attribute name should be concise and clear and use Chinese

Please return the JSON object directly without including other explanatory text

"""



Figure 7: Schema Extraction Prompt.

## D.2 QA Pair Construction

To create QA pairs, the following steps are followed: **Text Preprocessing:** The document is preprocessed to extract key entities, facts, and relations. This can include financial indicators, government policies, and medical terms. **Question Generation:** For each piece of information, a relevant question is formulated. For example, from a financial table, a question might be "What is the GDP growth rate for the year 2023?". **Answer Generation:** The answer is then generated by GPT-4.1, based on the content in the document.This process is repeated for each domain and table.

## D.3 Evaluation Metrics

The generated QA pairs are evaluated based on the following metrics:

- **Coverage:** The percentage of relevant information from the document captured by the QA pairs.

- **Format:** The degree to which the table generated by the model adheres to a consistent structure.

- **Reasoning:** The quality of logical connections made in the generated questions and answers.



# Schema Integration Prompt

"""

Please analyze multiple schemas in the following {domain} domain:

{schemas}

Please design a final schema with the following requirements:

1. Extract common attributes of all schemas, that is, attributes contained in each schema. For example, if attribute 1 can be found in the first schema but not in the second schema, then discard the attribute

2. Each attribute is a first-level title, that is, each is a separate field

3. Keep the fields contained in all schemas, and make sure that the fields of the final schema are contained in each schema

4. Ensure that the attribute names are unified and conform to the characteristics of the {domain} domain

5. Provide a clear description for each attribute

6. Ensure that the final schema attribute field is concise and can reflect the content of all articles and observe the similarities and differences

Please return the JSON object directly.

"""



Figure 8: Schema Integration Prompt.

- **Comprehension:** The ability of the model to correctly understand and answer questions based on the document's context.
- **Hallucination Control:** The model's ability to prevent the generation of false or unsupported information.

Each dimension is evaluated with a score from 0 to 1, where higher values indicate better performance.

## D.4 EXPERIMENTAL SETUP

The models compared in this study include:

- **GPT-4o:** The base model without any additional techniques for improving table generation.
- **GPT-4o (T3):** A model that uses the T3 method to extract article triples and form tables, capturing fine-grained information.

- **GPT-4o (DDST):** A model that employs DDST to select domain-specific gates and apply JSON Schema to optimize table generation.

Each model evaluates 12 QA pairs in four fields (finance, government, law, and healthcare), and the tables generated by different methods are answered using the big model, and then the answer results are compared on the above five dimensions.

## D.5 Results and Analysis

| Metric | GPT-4o | GPT-4o (T3) | GPT-4o (DDST) |
|---|---|---|---|
| **Coverage** | 0.47 | 0.67 | 0.79 |
| **Formatting** | 0.34 | 0.71 | 0.67 |
| **Reasoning** | 0.58 | 0.74 | 0.81 |
| **Compre** | 0.73 | 0.89 | 0.88 |
| **HC** | 0.41 | 0.74 | 0.83 |

Table 4: Comparison of T3 and DDST table results in five dimensions.Compre stands for Comprehension, HC stands for Hallucination Control.

The experimental results are summarized in Table 4. The table shows the scores for each model across the five evaluation dimensions.

From the results, it is observed that:**GPT-4o (DDST)** outperforms the other models in terms of coverage, reasoning, and hallucination control. This can be attributed to the DDST method's ability to select relevant domain gates and apply JSON Schema to capture more document information, particularly in the finance and government domains.**GPT-4o (T3)** excels in formatting and comprehension, benefiting from the triple extraction method that creates well-structured tables and captures fine-grained information.textbfGPT-4o serves as a baseline, with lower performance across all dimensions, as it lacks the additional techniques employed by the T3 and DDST methods.

In this study, we compared the performance of three table generation methods using the AUTO-QA framework. The results demonstrate that GPT-4o (DDST) provides superior coverage, reasoning, and hallucination control, making it the most effective method for generating high-quality QA pairs in complex domains. However, GPT-4o (T3) outperforms in terms of format and comprehension due to its fine-grained table structure extraction. These findings suggest that DDST and T3 methods complement each other, with each excelling in different aspects of the table generation task.

# E  Information Extraction Prompt

All the inputs used in our experiments to the large model prompt are as follows.

The first is to extract the corresponding table prompt directly from the original data, as shown in Figure 10, that is, multiple documents. The result is generated in column-first JSON format.

Next, we list all the prompts used in the DDST method, which consists of three parts in total.

The first step is to use prompt to determine the domain of the input document and generate a corresponding domain-specific prompt, as shown in Figure 11. For each piece of data, we will randomly select two articles from the original data and send them into the large model for domain judgment.

In the second step, we pass the large model to generate an intermediate json schema based on the generated domain-specific prompt words and related references, as shown in Figure 7,. We use the sliding window mechanism to generate a json schema for every four articles, and ensure that the generated json schema contains fields and their type restrictions.

In the third step, we integrate all generated schemas into a complete schema, ensuring that the schema can cover the information of all documents and the generated attribute fields meet the requirements, as shown in Figure 8.

Finally, we extract the appropriate table header fields and their constraints from the generated final schema, and use these header fields and constraints to find data from the original document and fill the final table, as shown in Figure 9.

## LIMITATIONS

Although the DDST method has achieved remarkable results in the multiple documents to table(Docs-to-Table) generation task, several limitations warrant future research. First, DDST's performance heavily relies on the language understanding and structured reasoning capabilities of the underlying foundation models. Some open-source models perform poorly in critical steps such as tuple extraction and logical deduction, limiting the advancement of the entire pipeline. Second, current experiments primarily focus on Fine-Tune and Zero-Shot settings, with no exploration of performance in Few-Shot scenarios. Constrained by context length and prompt design, this may affect its applicability in low-resource environments. Finally, there is still some semantic loss during information conversion—for example, certain detailed information is not correctly mapped to header fields or is misjudged as redundant content. Future research will focus on improving the system's automation, enhancing the model's structured reasoning capabilities, and exploring more effective information retention mechanisms to further enhance its robustness and generalization in complex real-world scenarios.

## ETHICS STATEMENT

The FGLM dataset and experiments in this work adhere to ethical research practices, utilizing publicly available sources from finance, government, law, and medicine domains, including financial reports, procurement announcements, court rulings, and pharmaceutical specifications, all compliant with transparency requirements. Personal information in legal and gov datasets was randomly replaced using named entity recognition (NER) techniques to ensure privacy. Data preprocessing maintained the integrity of the original sources, and experiments aligned with their intended research purpose of advancing multiple documents table generation. The dataset and code are open-sourced to promote transparency and, to the best of our knowledge, this work introduces no additional ethical risks or harms, adhering to data privacy and research ethics standards.

## Populate the table based on Schema

"""

You are a professional information extraction expert. Please extract the corresponding values from 10 articles according to the fields and field descriptions defined in the following JSON Schema.

Schema definition:

{schema}

Article content:

{articles}

Please extract information strictly according to the following requirements:

1. The returned field name must be in Chinese, and the corresponding value is extracted from each article according to the field type defined in the Schema. If it is not found, it is set to null

2. Strictly abide by the field type defined in the Schema:

- string type: extract text, if it exceeds 20 characters, it is set to null

- number type: extract numbers, retain the accuracy of the value

- array type: extract in array format

3. Make sure that the extracted value conforms to the description and meaning of the field

4. For fields of numeric type (such as EPS, revenue, etc.), pay attention to the consistency of units

5. Note that each field name returned must be in Chinese and correspond to 10 values, respectively from 10 articles

Please return the extraction results in JSON format, the format is as follows:

{{

"field name 1": ["value of article 1", "value of article 2", ..., "value of article 10"],

"field name 2": ["value of article 1", "value of article 2", ..., "value of article 10"],

...

}}

Please make sure the output is in valid JSON format and do not add any extra text.

"""

20

Figure 9: Populate the table based on Schema.

## Domain judgment and exclusive prompt word generation Prompt

- """

- Please analyze the following two articles

- {articles}

- Please complete the following tasks:

- 1. Determine the field to which the article belongs (financial/medical/legal/government/...)

- 2. Based on the content of the article, generate an expert prompt suitable for the field for subsequent JSON Schema generation

- Please return the result in JSON format, do not return other content, the format is as follows:

- {{

- "domain": "Field English name",

- "domain_prompt": "You are an expert in the {domain} field, please design a JSON Schema based on the following information. Documents in this field usually contain [domain characteristics], and special attention should be paid to [article topic]"

- }}

- Please ensure that domain_prompt contains the following content:

- 1. Domain characteristics: Describe the typical structure and content of documents in this field

- 2. Article topic: The common topic described by the two articles

- """

Figure 10: Direct Table Extraction Prompt.

## Domain judgment and exclusive prompt word generation Prompt

- """

- Please analyze the following two articles

- {articles}

- Please complete the following tasks:

- 1. Determine the field to which the article belongs (financial/medical/legal/government/...)

- 2. Based on the content of the article, generate an expert prompt suitable for the field for subsequent JSON Schema generation

- Please return the result in JSON format, do not return other content, the format is as follows:

- {{

- "domain": "Field English name",

- "domain_prompt": "You are an expert in the {domain} field, please design a JSON Schema based on the following information. Documents in this field usually contain [domain characteristics], and special attention should be paid to [article topic]"

- }}

- Please ensure that domain_prompt contains the following content:

- 1. Domain characteristics: Describe the typical structure and content of documents in this field

- 2. Article topic: The common topic described by the two articles

- """

Figure 11: Domain judgment and exclusive prompt word generation prompt.

# Government open procurement announcements

Project Overview

Potential bidders for the Yibin Natural Disaster Emergency Response Capacity Enhancement Project – Grassroots Disaster Prevention Project (Second Batch) (Fire Pump Rescue Category) should obtain the bidding documents from the Sichuan Provincial Government Procurement Integrated Platform Project Electronic Transaction System (hereinafter referred to as the "Project Electronic Transaction System") and submit the bidding documents before 10:00 AM on January 21, 2025 (Beijing Time).

I. Basic Information of the Project

- Project Number: N5115012024000618

- Project Name: Yibin Natural Disaster Emergency Response Capacity Enhancement Project – Grassroots Disaster Prevention Project (Second Batch) (Fire Pump Rescue Category)

- Procurement Method: Open Tender

- Budget Amount: RMB 6,040,613.00

- Procurement Requirements: See attached procurement requirements

- Contract Performance Period:

   Purchase Package 1: 90 days from the date of contract signing

- Are joint bids accepted?

   Purchase Package 1: Joint bids are not accepted

II. Qualification Requirements for Applicants

1. Must meet the provisions of Article 22 of the Government Procurement Law of the People's Republic of China;

2. Qualification requirements to implement government procurement policies:

   Purchase Package 1: None

3. Specific qualification requirements for this project:

   Purchase Package 1: None

III. Obtaining Bidding Documents

- Time: December 31, 2024 to January 7, 2025, daily from 00:00:00 to 12:00:00 and 12:00:00 to 23:59:59 (Beijing Time)

- Method: Log in to the Project Electronic Transaction System → Bidding (Response) Management → Select this project under "Not Obtained Procurement Documents" to get the tender documents

- Mode: Online acquisition

- Price: RMB 0 yuan

IV. Deadline for Submission of Bidding Documents, Opening Time and Location

- Submission Deadline: 10:00 AM on January 21, 2025 (Beijing Time)

- Submission Location: Submit bidding documents online via the Project Electronic Transaction System – Bidding (Response) Management

- Opening Location: Participate in the opening session through the Project Electronic Transaction System – Bidding / Opening Hall

V. Announcement Period At least 5 working days from the date of publication of this announcement.

VI. Other Supplementary Matters Supervision Department for This Procurement:

Yibin Municipal Finance Bureau

Tel: 0831-8228012

Address: No. 300 Yaowan Road, Nan'an West District, Yibin City

VII. Contact Information

1. Purchaser Information

   Name: Yibin Emergency Management Bureau  Address: No. 197 Nanguan Road, Xuzhou District, Yibin City  Contact: 0831-3883106

2. Procurement Agency Information Name: Yibin Municipal Government Procurement Center  Address: 4th Floor, Civic Center, Jinshajiang Avenue, Xuzhou District, Yibin City, Sichuan Province  Contact: 0831-8088817

3. Project Contact Person Name: Mr. Cao  Phone: 0831-3883106  ssued by: Yibin Government Procurement Center  Date: December 30, 2024

Figure 12: Government open procurement announcements.

## Government open procurement announcements

"ID": [1863298208936111153, 1863298208936111154, 1863298208936111155, 1863298208936111156, 1863298208936111157, 1863298208936111158, 1863298208936111159, 1863298208936111160, 1863298208936111161, 1863298208936111162], "Project Name": ["Fushun County 2024 Poverty Alleviation Population and Monitoring Object "Poverty Prevention Insurance" Service Procurement Project", "Public Security Network Risk Perception Equipment", "Suining Anju District No. 1 Primary School Off-campus Catering Delivery Service Procurement Project", "2024 Printer Consumables and Maintenance", "Sichuan University West China Hospital Liquid Nitrogen Supply System Procurement Project", "Teaching Integrate d Management System", "Litigation Document Service Self-service Terminal, Queuing System, Science and Technology Court Project", "2025 Information Operation and Maintenance Service Procurement", "Anzhou District High-tech Zone, Kaijiang Industrial Concentration Development Zone Regional Environmental Quality Routine Monitoring Fee", "Yingshan County 2025 Spring-2025 Fall Semester Compulsory Education Stage Nutrition Improvement School Canteen Raw Materials and Distribution Services"], "Procurement Unit": ["Fushun County Rural Revitalization Center", "Mianyang Public Security Bureau", "Suining Anju District No. 1 Primary School", "Chengdu Medical College First Affiliated Hospital", "Sichuan University West China Hospital", "Sichuan Prov incial People's Hospital", "Mianyang Fucheng District People's Court", "Sichuan Province Qionglai Prison", "Sichuan Mianyang Anzhou Industrial Park Management Committee", "Sichuan Province Nanchong Yingshan County Education and Sports Bureau"], "Announcement Time": ["2024/12/27 19:24", "2024/12/27 18:48", "2024/12/27 18:40", "2024/12/27 18:32", "2024/12/27 18:26", "2024/12/27 18:22", "2024/12/27 18:20", "2024/12/27 18:17", "2024/12/27 18:16", "2024/12/27 18:14"], "Get Bidding Document Time": [NaN, NaN, NaN, "December 30, 2024 to January 7, 2025\nEvery day from 00:00 am to 12:00??Afternoon:12:00 to 23:59 (Beijing time, except statutory holidays)", NaN, "December 30, 2024 to January 6, 2025\nEvery day from morning:00:00 to 12:00??Afternoon:12:00 to 23:59 (Beijing time, except statutory holidays)", NaN, NaN, NaN, "December 30, 2024 to January 6, 2025\nEvery day from morning:00:00 to 12:00??Afternoon:12:00 to 23:59 (Beijing time, except statutory holidays)"], "Price of bidding documents": [ NaN, NaN, NaN, "￥0", NaN, "￥0", NaN, NaN, NaN, "￥0"], "Where to obtain bidding documents": [NaN, NaN, NaN, "Project Electronic Trading System - Bidding (Response) Management - Select this project to obtain bidding documents in the procurement documents that have not been obtained", NaN, "Project Electronic Trading System - Bidding (Response) Management - Select this project to obtain bidding documents in the procurement documents that have not been obtained ", NaN, NaN, NaN, "Project Electronic Trading System - Bidding (Response) Management - Select this project to obtain bidding documents in the procurement documents that have not been obtained"], "Bid Opening Time": ["November 30, 2002 00:00", NaN, NaN, "January 21, 2025 10:30", NaN, "January 20, 2025 10:00", NaN, NaN, NaN, "January 20, 2025 09:30"], "Bid Opening Location": [NaN, NaN, NaN, "415, Block C, Dahecang, No. 511, Xingshi Road, Wuhou District, Chengdu", NaN, "Tender Opening Room, Room 2201-2203, 22F, Oriental Plaza, Block C, Zidong Road, Dongdajie, Jinjiang District, Chengdu", NaN, NaN, NaN, "Participate in the tender opening through the project electronic transaction system - tender opening/opening hall"], "Budget amount": ["See the announcement text for details", NaN, NaN, "￥120.000000 yuan (RMB)", NaN, "￥120.000000 yuan (RMB)", NaN, NaN, NaN, "￥2000.000000 yuan (RMB)"], "Project contact person": ["Yang Jianbo", "Pu Junxiu", "Ms. Li", "Ms. He", "Ni Xuelan, Wang Yu", "Mr. Yu, Ms. Zou, Ms. Li", "Liu Juan", "Ms. Jiang", "Ms. Huang", "Mr. Tang"], "Project contact number": ["0813-2306192", "Announcement 0816-2735361; Document preparation 0816-2335356; Tender opening and results 0816-2865208;", "0825-2225671", "028-85446608, 85445511, 85045522-8817", "13111881363", "028-65236981", "0816-2390739", "028-87088759, 86115713", "08166965888", "0817-2602338"], "Address of purchasing unit": ["No. 183 Haitang Road, Fushun County", "No. 298 West Section of Jiannan Road, Fucheng District", "No. 69 Zhengdong Street, Zhengdong Community, Rougang Office, Anju District, Suining City", "No. 278, Baoguang Avenue, Xindu District, Chengdu", "No. 37, Guoxue Lane, Chengdu", "No. 32, Section 2, West First Ring Road, Qingyang District, Chengdu, Sichuan Province", "No. 43, Mian'an Road, Fucheng District, Mianyang City", "Tutao Village, Linqiong Town, Qionglai City, Chengdu, Sichuan Province", "West Section of Wensheng Road, Anzhou District, Mianyang City, Sichuan Province", "No. 1, Fuxing 2nd Street, Yingshan County, Nanchong City, Sichuan Province"], "Contact information of the purchasing unit": ["13795553598", "13320891700", "Teacher Xiang 13882582930", "Teacher Huang; 028-83016299", "Teacher Zhang 028-85423272", "028-87393463", "15508006777", "Teacher Wu 028-88808053", "13320886702", "18681756038, contact person: Mr. Chen"], "Agency Name": ["Sichuan Huilong Bidding Consulting Co., Ltd.", "Mianyang Municipal Government Procurement Center", "Suining Chaochi Engineering Consulting Co., Ltd.", "Sichuan Wuzhou Bidding Agency Co., Ltd.", "Sichuan International Bidding Co., Ltd.", "Mianyang Chengrui Bidding Agency Co., Ltd.", "Sichuan Hongjie Bidding Agency Co., Ltd.", "Sichuan Yuanliang Bidding Agency Co., Ltd.", "Sichuan Zhongyi Bidding Agency Co., Ltd."], "Agency Address": ["No. 2 Danyang Street, Shuping Street, Ziliujing District, Zigong City, Sichuan Province (Purun Industrial Expo City Phase I Building 5) A5-3-Office 59", "3rd Floor, Municipal Government Service Center Building, No. 133 Mianxing East Road, Mianyang High-tech Zone", "C6 Building, Block 8, International Plaza, Suining Chepeilong Auto Department Store, Suining High-tech Zone, Suining City, Sichuan Province No. 203, 217, 218, 219, 220, 2nd Floor", "No. 415, Dahecang C Zone, No. 511, Xingshi Road, Wuhou District, Chengdu", "No. 1, 22nd Floor, Building 2, No. 66, Tianfu Fourth Street, High-tech Zone, Chengdu, China (Sichuan) Pilot Free Trade Zone", "Rooms 2201-2203, 22nd Floor, Building C, Oriental Plaza, Zidong Road, Dongdajie, Jinjiang District, Chengdu", "No. 1, 7th Floor, Unit 2, Building 16, Fulin Taohua Island, No. 1, North Section of Sanjiang West Road, Economic Development Zone, Mianyang, Sichuan", "A-E, 20th Floor, Guodong Central Business Building, No. 195, Shaanxi Street, Qingyang District, Chengdu", "No. 1516, Building A, Wanda Center, Economi c Development Zone, Mianyang, Sichuan", "No. 4, 17th Floor, Building 3, Financial Center, Jiangdong Middle Road, Gaoping District, Nanchong, Sichuan"], "Agent Contact Information": ["0813-2306192", "Announcement 0816-2735361; Document preparation 0816-2335356; Tender opening and results 0816-2865208;", "Ms. Li 0825-2225671", "Ms. He; 028-85446608, 85445511, 85045522-8817", "Ni Xuelan, Wang Yu 13111881363", "028-65236981", "0816-2390739", "Mr. He, Ms. Jiang 028-87088759, 86115713", "08166965888", "0817-2602338, Contact: Mr. Tang"]

Figure 13: Government Table Json.

## Legal public archives data

Civil Ruling of the People's Court of Faku County, Liaoning Province (2021) Liao 0124 Min Te 261 Applicant: Faku County Rural Credit Cooperative Union, unified social credit code: 912101247020716553, address: Henan Street, Faku Town, Faku County. Legal representative: Chen Dexu, chairman of the board of directors of Faku County Rural Credit Cooperative Union. Agent: Zuo Zhenyu, male, born on December 24, 1988, Han nationality, employee, address: Faku County, Liaoning Province. Applicant: Li Zehai, male, born on May 26, 1972, Han nationality, address: Kaiyuan City, Liaoning Province. Applicant: Yu Hong, female, born on November 19, 1972, Manchu nationality, address: Kaiyuan City, Liaoning Province. Applicant: Wang Anqi, female, born on November 21, 1994, Han nationality, address: Kaiyuan City, Liaoning Province. Attorney-in-fact: Wang Weizhu, male, born on September 21, 1962, Han nationality, address: Kaiyuan City, Liaoning Province. Applicant: Li Zhiyi, female, born on April 17, 1996, Han nationality, address: Kaiyuan City, Liaoning Province. Attorney-in-fact: Li Zehai, male, born on May 26, 1972, Han nationality, address: Kaiyuan City, Liaoning Province. On September 3, 2021, this court accepted the application of the applicants Faku County Rural Credit Cooperative Union, Li Zehai, Yu Hong, Wang Anqi, and Li Zhiyi for judicial confirmation of the mediation agreement, and conducted a review, which has now ended. The financial loan contract dispute between the applicants Faku County Rural Credit Cooperative Union, Li Zehai, Yu Hong, Wang Anqi, and Li Zhiyi was mediated by the Faku County Pre-trial People's Mediation Committee on September 3, 2021, and a mediation agreement was reached. As follows: 1. Applicants Li Zehai and Yu Hong shall repay the applicant Faku County Rural Credit Cooperative Union's loan interest of RMB 174,730.50, which shall be repaid before September 6, 2021; 2. If applicants Li Zehai and Yu Hong fail to repay the applicant Faku County Rural Credit Cooperative Union's loan interest on time, the proceeds from the discount, auction or sale of the real estate owned by Li Zhiyi (specifically: No. 1 (No. 7) Fajina European City, Faku County, house ownership certificate number: Shenfang Quanzheng Faku County No. ××, total construction area of 208.25 square meters) and the real estate owned by Wang Anqi (specifically: located at No. 1 (No. 8) Fajina European City, Faku County, house ownership certificate number: Shenfang Quanzheng Faku County No. ××, total construction area of 208.25 square meters) shall be used to repay first; 3. Applicants Wang Anqi and Li Zhiyi shall bear joint and several guarantee liability for the obligation to pay the above-mentioned funds. 4. The two parties have no other disputes. After review, this court believes that the mediation agreement reached by the applicants meets the statutory conditions for judicial confirmation of the mediation agreement. In accordance with Article 195 of the Civil Procedure Law of the People's Republic of China, the ruling is as follows: The mediation agreement reached by the applicants Faku County Rural Credit Cooperative, Li Zehai, Yu Hong, Wang Anqi, and Li Zhiyi on September 3, 2021, under the mediation of the Faku County Pre-trial People's Mediation Committee, is valid. The parties shall voluntarily perform their obligations in accordance with the terms of the mediation agreement. If one party refuses to perform or fails to perform in full, the other party may apply to the People's Court for compulsory execution. Judge: Ding Yi September 3, 2021 Clerk: Yang Chen

Figure 14: Legal public archives data.

## Law Table Json

"Case No.": ["(2021) Liao0404 Minchu 2915", "(2021) Liao0124 Minte 261", "(2021) Qing0105 Zhi 2127", "(2021) Liao0202 Zhi 5977-1", "(2021) Liao0112 Minchu 14917", "(2021) Liao0624 Zhi 1949", "(2021) Liao1002 Zhi 559-6", "(2021) Qing0224 Zhi 376", "(2021) Liao0214 Minchu 6881", "(2021) Liao02 Minzhong 7232"], "Case Name": ["Civil Ruling of the First Instance in the Civil Dispute over Property Service Contract between Ningbo Nasen Property Management Co., Ltd. and Fu Meiyu", "Civil ruling on the special procedure for confirming the validity of the people's mediation agreement applied by Faku County Rural Credit Cooperative Union and Li Zehai", "Execution notice on the execution of the loan contract dispute between the applicant: Li Qiuge and the person to be executed: Zhao Zenglu", "The first execution ruling on civil economy between the Dalian Branch of Ping An Bank Co., Ltd. and Pan Lirong", "The first instance civil ruling on the reputation rights dispute between Wei Shoulan, Luan Junxue and others", "The first execution notice on the contract, gratuitous management and unjust enrichment dispute between Yang Xueqin", "The first execution notice on the lease contract dispute between Meizhou Hanlong Engineering Construction Co., Ltd.", "The execution notice on the execution of the marriage and family dispute between the applicant Li", "The first instance civil ruling on the labor contract dispute between Guo Fuchun and Zhao Baogui", "The second instance civil ruling on the contract dispute between Linfeng Bamboo and Wood Trading Company and Teng Renping in the timber trading market in Ganjingzi District, Dalian"], "Court": ["Fushun Wanghua District People's Court", "Faku County People's Court of Liaoning Province", "Chengbei District People's Court of Xining City", "Zhongshan District People's Court of Dalian City", "Hunan District People's Court of Shenyang City", "People's Court of Kuandian Manchu Autonomous County, Liaoning Province", "People's Court of Baita District, Liaoyang City", "People's Court of Hualong Hui Autonomous County", "People's Court of Pulandian District, Dalian City", "Intermediate People's Court of Dalian City, Liaoning Province"], "Region": ["Fushun City", "Faku County, Liaoning Province", "Xining City", "Dalian City", "Shenyang City", "Kuandian Manchu Autonomous County, Liaoning Province", "Liaoyang City", "Hualong Hui Autonomous County", "Dalian City", "Dalian City, Liaoning Province"], "Case Type": ["Civil Cases", "Civil Cases", "Execution Cases", "Execution Cases", "Civil Cases", "Execution Cases", "Execution Cases", "Execution Cases", "Civil Cases", "Civil Cases"], "Case Type Code": [1, 1, 2, 2, 1, 2, 2, 2, 1, 1], "Trial Procedure": ["First Instance Civil Cases", "Special Procedures", "Execution Implementation", "Execution Implementation", "First Instance Civil Cases", "Execution Implementation", "Execution Implementation", "Execution Implementation", "First Instance Civil Cases", "Civil Second Instance"], "Judgment Date": ["2021-09-02", "2021-09-03", "2021-09-01", "2021-09-02", "2021-09-02", "2021-09-02", "2021-09-02", "2021-09-01", "2021-09-03", "2021-09-03"], "Publication Date": ["2021-09-04", "2021-09-04", "2021-09-03", "2021-09-03", "2021-09-04", "2021-09-03", "2021-09-04", "2021-09-03", "2021-09-04", "2021-09-04"], "Parties": ["Ningbo Naisen Property Management Co., Ltd.; Fu Meiyu", "Faku County Rural Credit Cooperative Union; Li Zehai; Yu Hong; Wang Anqi; Li Zhiyi", "Applicant for execution: Li Qiuge; Executed person: Zhao Zenglu", "Ping An Bank Co., Ltd. Dalian Branch; Pan Lirong", "Wei Shoulan; Luan Junxue; Beijing ByteDance Technology Co., Ltd.", "Yang Xueqin", "Meizhou Hanlong Engineering Construction Co., Ltd.", "Applicant for execution Li Mou", "Guo Fuchun; Zhao Baogui", "Linfeng Bamboo and Wood Business Store, Ganjingzi District Timber Trading Market, Dalian; Teng Renping"], "Cause": ["Property service contract disputes", "Application for confirmation of the validity of the people's mediation agreement", "Loan contract disputes", "Intellectual property rights, infringement disputes; civil", "Reputation disputes", "Contract, gratuitous management, unjust enrichment disputes", "Lease contract disputes", "Marriage and family disputes", "Labor contract disputes", "Contract disputes"], "Legal basis": ["Civil Procedure Law of the People's Republic of China: Article 154, paragraph 1; Civil Procedure Law of the People's Republic of China: Article 154, paragraph 1, item 1; Civil Procedure Law of the People's Republic of China: Article 154, paragraph 1, item 2; Civil Procedure Law of the People's Republic of China: Article 154, paragraph 1, item 3; Civil Procedure Law of the People's Republic of China: Article 154, paragraph 1, item 4; Civil Procedure Law of the People's Republic of China: Article 154, paragraph 1, item 5; Civil Procedure Law of the People's Republic of China: Article 154, paragraph 1, item 6; Civil Procedure Law of the People's Republic of China: Article 154, paragraph 1, item 7; Civil Procedure Law of the People's Republic of China: Article 154, paragraph 1, item 8; Civil Procedure Law of the People's Republic of China: Article 154, paragraph 1, item 9; Civil Procedure Law of the People's Republic of China: Article 154, paragraph 1, item 10; Civil Procedure Law of the People's Republic of China: Article 154, paragraph 1, item 11", "Civil Procedure Law of the People's Republic of China: Article 195", NaN, "Civil Procedure Law of the People's Republic of China: Article 242, paragraph 1; Civil Procedure Law of the People's Republic of China: Article 242, paragraph 2", "Civil Procedure Law of the People's Republic of China: Article 118, paragraph 1; Civil Procedure Law of the People's Republic of China: Article 118, paragraph 2; Civil Procedure Law of the People's Republic of China: Article 118, paragraph 3", NaN, NaN, NaN, "Civil Procedure Law of the People's Republic of China: Article 145, paragraph 1", "Civil Procedure Law of the People's Republic of China: Article 170, paragraph 1, item 3"]

Figure 15: Law Table Json.

1404
1405
1406
1407
1408
1409
1410
1411
1412
1413
1414
1415
1416
1417
1418
1419
1420
1421
1422
1423
1424
1425
1426
1427
1428
1429
1430
1431
1432
1433
1434
1435
1436
1437
1438
1439
1440
1441
1442
1443
1444
1445
1446
1447
1448
1449
1450
1451
1452
1453
1454
1455
1456
1457

# Medical instructions

Approval date: Revision date: Instructions for use of Lenalidomide Capsules Please read the instructions carefully and use under the guidance of a physician [Drug name] Generic name: Lenalidomide Capsules English name: Lenalidomide Capsules Chinese Pinyin: Lainadu'an Jiaonang [Ingredients] The main ingredient of this product is: Lenalidomide. Chemical name: 3-(4-amino-1-oxo-1,3-dihydro-2H-isoindol-2-yl)piperidine-2,6-dione Chemical structure: Molecular formula: $C_{13}H_{13}N_3O_3$ Molecular weight: 259.3 [Properties] The contents of this product are white or off-white powder. [Indications] This product is used in combination with dexamethasone to treat adult patients with multiple myeloma who have not been treated before and are not suitable for transplantation. This product is used in combination with dexamethasone to treat adult patients with multiple myeloma who have received at least one therapy. [Specification] 25mg [Pharmacokinetics] Absorption After oral administration of lenalidomide to healthy subjects under fasting conditions, this product is rapidly absorbed, and the plasma concentration reaches the highest within 0.5 to 1.5 hours after taking the drug. In patients and healthy subjects, the maximum plasma concentration (Cmax) and the area under the plasma concentration-time curve (AUC) can increase proportionally with the increase in dose. Multiple doses did not result in significant drug accumulation. The relative exposure of the S- and R-enantiomers of lenalidomide in plasma is approximately 56% and 44%, respectively. If healthy subjects receive high-fat and high-calorie foods at the same time, the degree of absorption will be reduced, resulting in a decrease in AUC of approximately 20% and a decrease in Cmax of 50%. However, in the key registration trials to establish the efficacy and safety of lenalidomide in the treatment of multiple myeloma, the fed state was not taken into account when administering the drug. Therefore, lenalidomide can be taken with food or on an empty stomach. Distribution In vitro, [14C]-lenalidomide has a low binding rate to plasma proteins, with an average binding rate of 23% and 29% in multiple myeloma and healthy subjects, respectively. After healthy subjects took lenalidomide 25 mg/day, lenalidomide could be detected in semen (the content was less than 0.01% of the dose taken), and the product could not be detected in semen after 3 days of discontinuation (see [Precautions]). Metabolism The results of human in vitro metabolism studies showed that lenalidomide is not metabolized by cytochrome P450 enzymes, suggesting that the simultaneous administration of lenalidomide and drugs that inhibit cytochrome P450 enzymes is unlikely to cause metabolic drug interactions in humans. In vitro studies have shown that lenalidomide has no inhibitory effect on CYP1A2, CYP2C9, CYP2C19, CYP2D6, CYP2E1, CYP3A or UGT1A1. Therefore, lenalidomide is unlikely to cause clinically significant drug interactions when used in combination with substrates of these enzymes. In vitro studies have shown that lenalidomide is not a substrate for: human breast cancer resistance protein (BCRP); multidrug resistance protein (MRT) transporters MRP1, MRP2, or MRP3; organic anion transporters (OAT) OAT1 and OAT3; organic anion transporting polypeptide 1B1 (OATP1B1); organic cation transporters (OCT) OCT1 and OCT2; multidrug and toxin extrusion protein (MATE) MATE1 and novel organic cation transporters (OCTN) OCTN1 and OCTN2. In vitro studies have shown that lenalidomide has no inhibitory effect on the bile salt export pump (BSEP), BCRP, MRP2, OAT1, OAT3, OATP1B1, OATP1B3, or OCT2. Lenalidomide also does not inhibit the formation of bilirubin glucuronide in human liver microsomes with UGT1A1 genotypes of UGT1A1*1/*1, UGT1A1*1/*28, and UGT1A1*28/*28. The plasma half-life is approximately 3 hours in healthy volunteers over the 5-25 mg/day dose range; 3 to 5 hours in patients with multiple myeloma, myelodysplastic syndrome, or mantle cell lymphoma. Excretion Lenalidomide is primarily excreted via the urinary tract. In subjects with normal renal function, renal excretion accounts for 90% of total clearance, and 4% of lenalidomide is excreted in the feces. Lenalidomide is minimally metabolized, with 82% of the dose excreted unchanged in the urine. Hydroxyl lenalidomide and N-acetyl lenalidomide account for 4.59% and 1.83% of the excretion, respectively. Renal clearance of lenalidomide exceeds the glomerular filtration rate, so the drug is at least somewhat actively secreted. Special Populations Pediatric Patients No pharmacokinetic data are available for lenalidomide in patients under 18 years of age. Elderly Patients No dedicated clinical studies have been conducted to evaluate the pharmacokinetics of lenalidomide in the elderly. The population pharmacokinetic analysis included patients aged 39 to 85 years and showed that age had no effect on the clearance (plasma exposure) of lenalidomide. Because elderly patients are more likely to have decreased renal function, caution should be exercised when selecting doses and renal function should be monitored. Patients with renal impairment The pharmacokinetics of lenalidomide were studied in U.S. patients with renal impairment due to non-malignant causes. In this study, 5 patients with mild renal impairment (CLcr 56-74 mL/min), 6 patients with moderate renal impairment (CLcr 33-46 mL/min), 6 patients with severe renal impairment (CLcr 17-29 mL/min), and 6 patients with end-stage renal disease requiring dialysis received a single oral dose of 25 mg lenalidomide. The control group consisted of 7 healthy subjects of similar age with normal renal function (CLcr 83-145 mL/min) who also received a single oral dose of 25 mg lenalidomide. The results of this study showed that the pharmacokinetic characteristics of lenalidomide in patients with mild renal impairment were similar to those of healthy subjects. The half-life of patients with moderate to severe renal impairment was extended by 3 times, while the total clearance was reduced by 66% to 75% compared with healthy subjects. The half-life of patients requiring hemodialysis was extended by about 4.5 times, and the total clearance was reduced by 80% compared with healthy subjects. Patients with renal impairment can clear approximately 30% of the drug from the body after 4 hours of dialysis. The recommended dose adjustment for patients with renal impairment is described in detail in [Dosage and Administration]. Patients with hepatic impairment Population pharmacokinetic analysis included a patient population with mild hepatic impairment (N = 16, total bilirubin > 1.0≤1.5 × ULN or AST > ULN), and the results showed that mild hepatic impairment did not affect the distribution of lenalidomide in the body. There are no data on patients with moderate to severe hepatic impairment. Other factors affecting pharmacokinetics Population pharmacokinetic analysis showed that body weight (33-135 kg), gender, race and different types of malignant blood tumors did not affect the clearance of lenalidomide in adult patients. Pharmacokinetics in Chinese patients with multiple myeloma A pharmacokinetic study was conducted on 11 Chinese patients with refractory/relapsed multiple myeloma who received a daily dose of 25 mg of lenalidomide. The plasma concentration of lenalidomide reached its peak approximately 1 hour after administration, and the average terminal half-life was approximately 3 hours. The average plasma exposure levels (Cmax and AUC) observed in Chinese patients were similar to historical data obtained from Caucasian patients. [Storage] Sealed and stored at room temperature (10-30°C). [Packaging] Double aluminum packaging (polyamide/aluminum/polyvinyl chloride cold-stamped solid pharmaceutical composite hard tablets, pharmaceutical aluminum foil). (1) 7 tablets/plate: 1 plate/box, 3 plates/box; (2) 10 tablets/plate: 2 plates/box, 3 plates/box, 4 plates/box. [Validity period] 36 months [Implementation standard] [Approval number] National Medicine Standard H20193006. [Manufacturer] Company name: Zhengda Tianqing Pharmaceutical Group Co., Ltd. Production address: No. 8, Julong North Road, Xinpu District, Lianyungang City Postal code: 222006 Telephone number: 0518-85804002 Fax: 0518-85806524 Website: Health consultation hotline: 400-728-5028 [Marketing authorization holder] Company name: Zhengda Tianqing Pharmaceutical Group Co., Ltd. Company address: No. 369, Yuzhou South Road, Lianyungang City, Jiangsu Province "

Figure 16: Medical instructions.

## Medical Table Json

"Active ingredients": ["Lenalidomide", "Metformin Hydrochloride", "Candesartan Cilexetil", "Ibuprofen", "Amlodipine Besylate", "Metformin Hydrochloride", "Finasteride", "Cefuroxime Axetil", "Montmorillonite", "Cefalexin"], "Active ingredients (English)": ["Lenalidomide", "Metformin Hydrochloride", "Candesartan Cilexetil", "Ibuprofen", "Amlodipine Besylate ", "Metformin Hydrochloride", "Finasteride ", "Cefuroxime Axetil", "Montmorillonite", "Cefalexin"], "Drug name": ["Lenalidomide Capsules", "Metformin Hydrochloride Tablets", "Candesartan Cilexetil Tablets", "Ibuprofen Granules", "Amlodipine Besylate Tablets", "Metformin Hydrochloride Tablets", "Finasteride Tablets", "Cefuroxime Axetil Tablets", "Montmorillonite Powder", "Cefalexin Capsules"], "Drug Name (English)": ["Lenalidomide Capsules", "Metformin Hydrochloride Tablets", "Candesartan Cilexetil Tablets", "Ibuprofen Granules", "Amlodipine Besylate Tablets", "Metformin Hydrochloride Tablets", "Finasteride Tablets", "Cefuroxime Axetil Tablets", "Montmorillonite Powder", "Cefalexin Capsules"], "Dosage Form": ["Capsules", "Tablets", "Tablets", "Granules", "Tablets", "Tablets", "Tablets", "Tablets", "Powders", "Capsules"], "Route of Administration": ["Oral", "Oral", "Oral", "Oral", "Oral", "Oral", "Oral", "Oral", "Oral", "Oral", "Oral", "Oral"], "Specifications": ["25mg", "0.25g", "4mg", "0.2g", "5mg (calculated as C20H25ClN2O5)", "0.25g", "5mg", "calculated as C16H16N4O8S 0.5g", "Each bag contains 3 grams of montmorillonite", "calculated as C16H17N3O4S 0.25g"], "Reference preparation": ["No", "No", "No", "No", "No", "No", "No", "No", "No", "No"], "ATC code": ["L04AX04", "A10BA02", "C09CA06", "M01AE01", "C08CA01", "A10BA02", "G04CB01", "J01DC02", "A07BC05", "J01DB01"], "Approval number/registration certificate number": ["National Medicine Standard H20193006", "National Medicine Standard H37020550", "National Medicine Standard H20143260", "National Medicine Standard H20066208", "National Medicine Standard H20066843", "National Medicine Standard H20184130", "National Medicine Standard H20051983", "National Medicine Standard H20067966", "National Medicine Standard H20093622", "National Medicine Standard H13020791"], "Approval Date": ["2019/10/21", "2019/9/10", "2019/10/21", "2019/10/18", "2019/10/18", "2019/10/21", "2019/9/18", "2019/9/30", "2019/10/18", "2019/10/23"], "Marketing Authorization Holder": ["Chia Tai Tianqing Pharmaceutical Group Co., Ltd.", NaN, NaN, "CSPC Pharmaceutical Group Ouyi Pharmaceutical Co., Ltd.", NaN, NaN, NaN, "Sinopharm Group Shantou Jinshi Pharmaceutical Co., Ltd.", NaN, "CSPC Pharmaceutical Group Ouyi Pharmaceutical Co., Ltd."], "Manufacturer": ["Chia Tai Tianqing Pharmaceutical Group Co., Ltd.", "Penglai Nuokang Pharmaceutical Co., Ltd.", "Tiandi Hengyi Pharmaceutical Co., Ltd.", "CSPC Pharmaceutical Group Ouyi Pharmaceutical Co., Ltd.", "Yichang Dongyangguang Changjiang Pharmaceutical Co., Ltd.", "Harbin Pharmaceutical Group Pharmaceutical Factory No. 6", "China Resources Secco Pharmaceuticals Co., Ltd.", "Sinopharm Group Shantou Jinshi Pharmaceutical Co., Ltd.", "Beijing Hanmei Pharmaceutical Co., Ltd.", "CSPC Pharmaceutical Group Ouyi Pharmaceutical Co., Ltd."]

Figure 17: Medical Table Json.

