# OpenReview forum: "Docs2Table: Bridging Multiple Documents and Structured Tables via LLMs"
_ICLR.cc/2026/Conference — ICLR 2026 Conference Withdrawn Submission_

### Official Review · Reviewer_Ak8A · 2025-10-20

**Soundness:** 2
**Presentation:** 1
**Contribution:** 2
**Rating:** 2
**Confidence:** 4

**Summary:**

This paper introduces Docs2Table, a novel information extraction task focused on generating structured tables from multiple documents by synthesizing and comparing their content. To facilitate this, the authors have developed FGLM, the first multi-domain benchmark dataset covering finance, government, law, and medicine, sourced from real-world scenarios. They also propose a two-stage pipeline method named DDST (Docs-Domain-Schema-Table), which incorporates a domain gating mechanism to identify the document's domain and uses a JSON Schema as an intermediate step to define the table's structure and constraints. Experimental results show that the DDST method significantly outperforms existing models on both traditional datasets and the new FGLM benchmark, demonstrating strong generalization capabilities across different domains.

**Strengths:**

1. The paper introduces the novel and challenging task of Docs2Table, which focuses on converting multiple documents into a structured table.
2. The authors have contributed FGLM, a new Docs2Table dataset that encompasses multiple domains.

**Weaknesses:**

1. Ambiguity in Dataset Construction: The description of FGLM's construction is overly brief, which raises concerns about its quality and reproducibility. Several key details are missing from both the main paper and the appendix:
   - Document Provenance: The paper states that the documents were sourced from the web (line 205) but provides no specific details about the websites or the crawling methodology.
   - Annotation Protocol: The annotation process is only vaguely described (lines 203-207). A detailed annotation guideline is crucial, as different annotators may have varying interpretations of the documents and criteria for extracting important information, potentially introducing bias. The paper should specify the standards established to ensure annotation consistency.
   - Dataset Statistics: The paper lacks basic statistics for the FGLM dataset, such as the total number of tables and the average number of rows and columns.
   - Annotator Details: The annotators are described simply as "domain experts" (line 207). The paper would benefit from providing more specific information, such as the number of annotators, their recruitment criteria, and the average time taken for annotation.
   - Document Similarity Metric: The use of TF-IDF for measuring document similarity appears overly simplistic. This method relies solely on lexical overlap and may fail to capture deeper semantic relationships and domain-specific information [1].
2. Generalizability of the Domain Gating Mechanism: I have reservations about the generalization capability of the domain gating mechanism in the proposed DDST model.
   - This component requires Supervised Fine-Tuning (SFT) of a Large Language Model (LLM), yet the authors do not detail this fine-tuning process. It is unclear whether the data domains used for SFT are disjoint from those used for testing.
   - This lack of information casts doubt on whether the model can accurately identify domains and generate appropriate keywords when encountering a domain unseen during the SFT phase.
   - Consequently, the comparisons in Tables 2 and 3 may be unfair, as most of the baseline models and methods evaluated are zero-shot approaches, whereas the proposed method has been fine-tuned on in-domain data.

[1] Text Document Clustering: Wordnet vs. TF-IDF vs. Word Embeddings

**Questions:**

- Minor Errors:
  - The heading for the first section appears to be incorrect; it should be "1. Introduction".
  - The citation formatting throughout the manuscript does not seem to follow a standard style.
  - The paper is missing a "Related Work" section, which is essential for contextualizing the contribution.
  - There is a discrepancy between Figure 3, which illustrates four steps, and the description on line 250, which states there are only three. Step 1 in the figure is not mentioned in the text, causing confusion.
- Suggestions for Improvement:
  - I recommend reducing the use of abbreviations in the tables to improve readability.
  - I suggest moving the results for the baselines T3 and DDSHT from Table 3 (Ablation Study) to the main results in Table 2. As these are methods from prior work, they should be compared directly with the main proposed model rather than being presented as part of an ablation analysis.

---

### Official Review · Reviewer_4pcT · 2025-10-21

**Soundness:** 2
**Presentation:** 2
**Contribution:** 3
**Rating:** 2
**Confidence:** 4

**Summary:**

This paper proposes **Docs2Table**: aggregating multiple related documents into a single structured table for real-world domains such as finance, government, law, and medicine. The authors build a cross-domain benchmark **FGLM** (long inputs, dynamic headers, expert gold labels) and introduce a two-stage method **DDST**: first domain identification and schema induction, then schema-constrained extraction and population to reduce hallucinations and improve consistency. Experiments show that, compared with end-to-end and existing text-to-table methods, DDST significantly improves header and cell accuracy, and thinking-style models also benefit. Overall, the work advances reliable multi-document–to–table generation via aligned task, dataset, and method.

**Strengths:**

1. The motivation is solid and practically valuable: in real-world settings we often need to process **multiple documents** jointly; a **multi-document Docs2Table** dataset is therefore warranted.
2. The experimental design is comprehensive, covering a broad set of models and adequately validating the authors’ claims.

**Weaknesses:**

1. The dataset construction may diverge from real scenarios—for example, each instance is fixed to **10 documents**, and **relevant-document selection** relies on **TF-IDF** rather than manual curation or stronger retrieval methods.
2. **DDST** appears to have **limited generalizability**: its **Domain Gating** depends on a predefined domain set, which may be hard to specify in practice, and the **sliding-window** procedure could introduce significant efficiency overhead.

Despite the valuable motivation, the current method and writing issues are substantial; I recommend **reject**.

**Questions:**

1. Why use **TF-IDF** instead of an encoder-based retriever (e.g., **DPR** [1]) for selecting relevant documents?
2. The paper has numerous formatting problems, including but not limited to:
   * The **Section 1** title is incorrect.
   * Citation **formatting** is inconsistent.
   * The **Evaluation Metrics** section is confusingly organized.
   * **Figures 2–4** contain excessive white space, making the text appear sparse.
   * **Appendix A** is incorrect.

[1] Dense Passage Retrieval for Open-Domain Question Answering. Karpukhin, et al. EMNLP 2020.

---

### Official Review · Reviewer_Weab · 2025-10-30

**Soundness:** 2
**Presentation:** 1
**Contribution:** 2
**Rating:** 0
**Confidence:** 4

**Summary:**

This paper introduces a novel task of extracting structured tables from multiple documents without predefined schemas. The key challenge lies in dynamically defining the table columns by comparing and summarizing information across documents. The authors construct a dataset for this task (apparently comprising 1,802 documents, though there are inconsistencies in the description) and propose a two-stage LLM-based approach: first generating and refining table schemas per document, then extracting table rows based on the finalized schema.

**Strengths:**

1. Proposing a new task: schema-agnostic table extraction from multiple documents.
2. Curating a dataset and conducting experiments to validate the proposed method.

**Weaknesses:**

1. Data Annotation and Quality: The paper lacks details on how the dataset was annotated. It is mentioned that documents and tables are sourced from real-world websites and corrected by domain experts (line 205), but the annotation process, quality control, and inter-annotator agreement are not discussed. This raises concerns about data reliability.
2. Task Definition and Completeness: The task of schema-agnostic table extraction lacks clarity on how "completeness" of columns is defined. The paper does not address scenarios with multiple valid predictions or how annotation "completeness" is ensured. Furthermore, the motivation for this task is unclear. In real-world applications, data extraction is often driven by specific analytical needs, so the columns to extract are typically predefined. The authors should justify the necessity of schema-free extraction with concrete use cases.
3. Presentation Issues: I will list only some examples here:
  - Figures: Figures 1 and 2 are overly large yet provide limited useful information.
  - Citations: Inconsistent citation formatting (e.g., mixing \citet and \citep).
  - Data Inconsistencies: The document count is ambiguous. Table 1 describes 1,802 as the "Document Number," but line 195 states that "FGLM contains 1,802 entries, with each entry comprising 10 documents."
  - Language Quality: The English in Appendix line 676 is poorly phrased.
  - Typos: Appendix Figure 16: The title "medical instruction" is unclear and irrelevant to the context.

**Questions:**

Will the dataset and evaluation code be publicly released to facilitate reproducibility and future research?

---

### Official Review · Reviewer_bJaK · 2025-11-01

**Soundness:** 3
**Presentation:** 2
**Contribution:** 3
**Rating:** 6
**Confidence:** 4

**Summary:**

This paper introduces "Docs2Table," a novel and challenging task for structured information extraction. The authors argue that existing Text-to-Table research, which primarily focuses on single-document scenarios, fails to address the real-world need for comparative analysis across multiple, domain-specific documents.

The paper makes three primary contributions:

1. **A new task (Docs2Table):** Formally defines the task of generating a single, structured table from a collection of related documents, requiring cross-document comparison and domain-specific schema adherence.

2. **A new benchmark (FGLM):** Introduces a large-scale, multi-domain (Finance, Government, Law, Medicine) dataset derived from real-world documents to support this task.

3. **A new method (DDST):** Proposes a two-stage pipeline, Docs-Domain-Schema-Table, which uses a domain-gating mechanism to identify the document domain, followed by a multi-step process to generate an intermediate JSON Schema, which finally constrains a large language model (LLM) during table population.

Experimental results on FGLM demonstrate that the proposed DDST method, when applied to SOTA LLMs, outperforms zero-shot baselines.

**Strengths:**

1. **Significant Problem Formulation:** The paper's primary strength is its identification and formalization of a valuable, practical, and underserved problem. The critique of existing single-document benchmarks is accurate, and the Docs2Table task, which emphasizes domain-specific, comparative analysis from multiple long-form texts, is a significant and necessary step forward for the field.

2. **Valuable Dataset Contribution:** The FGLM dataset is a major contribution. It provides the community with the first large-scale, real-world benchmark for this new task. Its construction from diverse and complex domains (finance, law, etc.) and its use of lengthy, realistic documents (avg. 9,734 tokens) present a substantial challenge that will likely drive future research.

3. **Emphasis on Domain-Specificity:** The paper correctly identifies that structured data generation is not a one-size-fits-all problem. The explicit focus on domain adaptation (e.g., in the FGLM dataset design and the DDST's "domain gate" mechanism) is a crucial insight.

4. **Thorough Experimental Baseline:** The experiments are comprehensive, providing a solid set of baselines on the FGLM dataset by testing a wide variety of modern LLMs (fine-tuned, zero-shot, CoT) and evaluation metrics.

**Weaknesses:**

1. **Limited Methodological Novelty:** The proposed DDST method is less a novel architecture and more a complex, multi-stage prompt-chaining pipeline. This "method" consists of at least four separate sequential LLM calls (domain-gating, schema-extraction-per-window, schema-integration, and table-population). This approach is computationally expensive, highly brittle (error propagation is a major risk), and appears over-engineered for this specific task, offering limited generalizable insights for the field.

2. **Mismatch Between Claims and Task Definition:** There is a significant discrepancy between the paper's claims of "cross-document synthesis" and the formal task definition (Section 2.2). The task is defined as mapping each document $d_j$ to a single row $r_j$ in the output table. This 1-to-1 mapping does not inherently require the model to perform complex synthesis, information fusion, or contradiction resolution across documents. The task, as defined, more closely resembles a batch single-document extraction and normalization problem. The "comparative analysis" seems to be a downstream cognitive task for the human user reading the table, not a capability being evaluated in the model itself.

3. **Insufficient Rigor in Dataset Construction:** For a paper whose primary contribution is a new benchmark, the description of its construction lacks scientific rigor. The paper claims the data was "corrected by domain experts," but this is insufficient. There is no information on the expert guidelines, the annotation/correction protocol, or (most critically) any Inter-Annotator Agreement (IAA) metrics. Without IAA, the quality, consistency, and objectivity of the "gold standard" tables are unverifiable.

**Questions:**

1. Could the authors clarify the apparent discrepancy between the claim of "cross-document synthesis" and the 1-to-1 document-to-row task definition? Does the task, as evaluated, truly require synthesis, or is it primarily a batch extraction task? If it is the latter, the claims should be toned down.

2.  Can the authors please provide more details on the FGLM construction? Specifically:
a) What protocol or guidelines were given to the domain experts for correction?
b) Was a measure of Inter-Annotator Agreement (IAA) calculated? If so, what was it? If not, how can the community be assured of the dataset's consistency and objectivity?

3.  Given the fragility of the 4-stage DDST pipeline, have the authors evaluated a simpler, end-to-end baseline? For example, prompting a model like GPT-4o with all 10 documents and a single, carefully crafted prompt (perhaps with a few-shot example) to generate the final JSON directly? It is unclear if the complexity of DDST is necessary or if it simply overfits the FGLM dataset.

---

### Note · Authors · 2025-12-01

I have read and agree with the venue's withdrawal policy on behalf of myself and my co-authors.